Transcriptome-wide and expression analysis of the NAC gene family in pepino (Solanum muricatum) during drought stress

Yang Shipeng 1
Zhu Haodong 2
Huang Liping 1
Zhang Guangnan 1
Wang Lihui 1
Jiang Xiaoting 3
Zhong Qiwen 1 13997135755@163.com
1 Qinghai Key Laboratory of Vegetable Genetics and Physiology, Agriculture and Forestry Sciences Institute of Qinghai University, Qinghai University , Xining , P.R. China
2 Qinghai University , Xining , P.R. China
3 Qinghai Higher Vocational & Technical Institute, Ledu, P.R. China , Xining , China
Coupland George
Electronic publication date: 2021 Mar 29
Publication date: 2021
Volume: 9
Electronic Location ID: e10966
Received 2020 Jun 23; Accepted 2021 Jan 28
Copyright: © 2021 Yang et al.
Copyright year: 2021
Copyright holder: Yang et al.
License: This is an open access article distributed under the terms of the Creative Commons Attribution License, which permits unrestricted use, distribution, reproduction and adaptation in any medium and for any purpose provided that it is properly attributed. For attribution, the original author(s), title, publication source (PeerJ) and either DOI or URL of the article must be cited.
License URL: https://creativecommons.org/licenses/by/4.0/

Keywords: NAC, Drought stress, Solanum muricatum, Gene expression, Gene regulation

Funding: Association of Qinghai Science and Technology 2019QHSKXRCTJ06 Key Laboratory Project of Qinghai Science & Technology Department 2020-ZJ-Y02 Qinghai Agriculture and Forestry Science Innovation Fund 2020-NKY-02 This research was supported by the Association of Qinghai Science and Technology (2019QHSKXRCTJ06), the Key Laboratory Project of Qinghai Science & Technology Department (2020-ZJ-Y02), and the Qinghai Agriculture and Forestry Science Innovation Fund (2020-NKY-02). The funders had no role in study design, data collection and analysis, decision to publish, or preparation of the manuscript.

==============================
Solanum muricatum (Pepino) is an increasingly popular solanaceous crop and is tolerant of drought conditions. In this study, 71 NAC transcription factor family genes of S. muricatum were selected to provide a theoretical basis for subsequent in-depth study of their regulatory roles in the response to biological and abiotic stresses, and were subjected to whole-genome analysis. The NAC sequences obtained by transcriptome sequencing were subjected to bioinformatics prediction and analysis. Three concentration gradient drought stresses were applied to the plants, and the target gene sequences were analyzed by qPCR to determine their expression under drought stress. The results showed that the S. muricatum NAC family contains 71 genes, 47 of which have conserved domains. The protein sequence length, molecular weight, hydrophilicity, aliphatic index and isoelectric point of these transcription factors were predicted and analyzed. Phylogenetic analysis showed that the S. muricatum NAC gene family is divided into seven subfamilies. Some NAC genes of S. muricatum are closely related to the NAC genes of Solanaceae crops such as tomato, pepper and potato. The seedlings of S. muricatum were grown under different gradients of drought stress conditions and qPCR was used to analyze the NAC expression in roots, stems, leaves and flowers. The results showed that 13 genes did not respond to drought stress while 58 NAC genes of S. muricatum that responded to drought stress had obvious tissue expression specificity. The overall expression levels in the root were found to be high. The number of genes at extremely significant expression levels was very large, with significant polarization. Seven NAC genes with significant responses were selected to analyze their expression trend in the different drought stress gradients. It was found that genes with the same expression trend also had the same or part of the same conserved domain. Seven SmNACs that may play an important role in drought stress were selected for NAC amino acid sequence alignment of Solanaceae crops. Four had strong similarity to other Solanaceae NAC amino acid sequences, and SmNAC has high homology with the Solanum pennellii. The NAC transcription factor family genes of S. muricatum showed strong structural conservation. Under drought stress, the expression of NAC transcription factor family genes of S. muricatum changed significantly, which actively responded to and participated in the regulation process of drought stress, thereby laying foundations for subsequent in-depth research of the specific functions of NAC transcription factor family genes of S. muricatum.

Introduction

Solanum muricatum is a perennial plant of the nightshade family, Solanaceae, native to the Andes in South America. Compared with most fruits, S. muricatum is rich in potassium, vitamin C and selenium (Kola, 2010). The flesh is delicious and it has fragrance and succulence. Antioxidants in S. muricatum have been found to have medicinal value, such as lowering blood pressure, diuresis, preventing alcoholic fatty liver and anti-cancer properties (Cheng-Chin et al., 2011; Kola et al., 2015; Shathish & Guruvayoorappan, 2014; Shathish, Sakthivel & Guruvayoorappan, 2015). S. muricatum is now cultivated around the world and is being gradually recognized as an important medicinal plant. At present, China has planted a large number of S. muricatum in Qinghai, Gansu, Yunnan and other places, growing in high-altitude areas around 1,000–3,000 m above sea level. Abiotic stress factors such as high temperature, drought (Langeroodi et al., 2020; Zamarud et al., 2020), and biotic factors such as diseases and insect pests (Alcaide et al., 2020; Tadashi & Ken, 2019) have caused serious damage to the plant growth, development and production. These stress factors restrict the development and output in the industries dealing with S. muricatum.

Solanum muricatum is suitable for planting in relatively arid and barren areas; Jiuquan, Wuwei, and other places in the main planting areas of West Gansu are all Gobi desert areas. The main cultivation area of Yunnan contains the Stone Forest, part of the karst landform, characterized by rugged geology and poor soil nutrients (Tadashi & Ken, 2019). The frequent occurrence of extreme climates around the world has exacerbated the adverse effects of abiotic stress on agricultural production, especially drought stress. The primary stress effect caused by drought is high osmotic stress, usually referred to as osmotic stress. The secondary effects of drought stress are more complex and, although these may eventually lead to plant metabolic disorders, the most important consequence of drought stress is the accumulation of the plant hormone ABA as a result of osmotic stress; that, in turn, affects the adaptive response of the plant. Studies have shown that regulation of the NAC transcription factor family under drought stress enhances the osmotic tolerance ability of the plant “store” and senescent leaves that have lost their osmotic tolerance which act as “sources”. In this way, water in the plant is preferentially transported to the “store” to promote plant survival and the continuation of the species (Chen et al., 2018; Jinxiang et al., 2016). Besides this, NAC family genes have been shown to regulate a wide range of developmental processes, including fiber development (Shim et al., 2018), cell wall synthesis (Huang et al., 2015), cell expansion (Pei et al., 2013; Yunjun et al., 2014), leaf senescence and fruit ripening (Xuemin et al., 2018). Further, plant NAC transcription factors also play an important regulatory role in response to adverse stress conditions including drought (Yuan et al., 2019). It has been reported that alterations in the expression of NAC transcription factors can increase the plant’s tolerance to abiotic stress (Yong, Zhang & Lyu, 2019).

NAC transcription factors, as a plant-specific superfamily that is involved in various developmental processes, including stress response and tolerance (Kazuo et al., 2012), have been intensively investigated in recent years. The number of NAC transcription factors varies between 30 and 200 among different crops. There are 33 transcription factors in Physcomitrella patens (Rakhi & Swarnendu, 2019), 63 in coffee (Coffea canephora) (Dong et al., 2019; Mao et al., 2016), 87 in sesame (Sesamum indicum) (Zhang et al., 2018), 104 in pepper (Capsicum annuum L.) (Weiping et al., 2018) and 114 in Betula pendula (Shen et al., 2019; Weiping et al., 2018). There has been a great deal of research on the NAC in Solanaceae crops. In the tomato (Solanum lycopersicum), the NAC family member JUB1 interacts directly with the promoters of SlDREB1, SlDREB2 and SlDELLA, increasing the drought tolerance of tomato. JUB1 acts as a transcriptional regulator of drought tolerance and controls the regulatory network of abiotic stress-related genes, which is fairly conserved among Arabidopsis thaliana and tomato (Alshareef et al., 2019). Similarly, the NAC transcription factors SlNAC35 and SlNAC11 are involved in the response to drought stress and salt stress in tomato. Overexpression of SlNAC35 has shown that it can affect the growth and development of the roots, as well as promoting the expression of members of the ADP ribosylation factor (ARF) family, indicating that NAC have a role in mediating auxin conduction and regulation (Wang et al., 2016, 2017). In the potato (Solanum tuberosum L.), StNAC002, StNAC025, StNAC087 and StNAC091 enhance resistance while several NAC can enhance the disease resistance of potato to anti-phytophthora (Singh et al., 2013). There are 176 NAC (NtNAC) transcription factors in tobacco (Nicotiana tabacum L.), the earliest crop to be used for conducting functional studies of the NAC family, although there have been few conclusive reports (Tweneboah & Oh, 2017). NtNACR1 increases lateral root development and nicotine content during tobacco topping, while NtNAC2 increases the salt resistance of plants (Han et al., 2014). Compared to tomato and other Solanaceae crops, NAC proteins in pepper have somewhat longer sequences. CaNAC2 is mainly expressed in seeds and roots. The silencing of CaNAC2 verifies that this gene has a positive regulatory effect on the plants during cold and salt stress (Wei-Li et al., 2015).

Because of their prominent role in plant abiotic stress, growth and development, the NAC gene family is an important focus of research. S. muricatum is similar to the other solanaceous crops originating from the Andes. However, although other solanaceous have been intensively studied, the research on S. muricatum is only about 30 years old. This research shows that S. muricatum has high stress-resistance and excellent planting prospects in saline-alkaline and arid areas (Duman & Sivaci, 2015; Jinxiang et al., 2016; Prohens, Ruiz & Nuez, 2002). The mechanism of S. muricatum’s resistance against drought has not been investigated. The use of biotechnology to regulate a plant’s NAC transcription factors could enhance its resistance to adverse environments and would be of great significance for production. The exploration and analysis of the NAC gene family can provide technical support for S. muricatum breeding and cultivation. In this study, the NAC transcription factor family genes of S. muricatum were identified and analyzed, investigating their evolutionary classification, protein physicochemical properties, conserved domains, tissue-specific expression and expression of conserved domain genes under different degrees of drought stress. The study aims to provide a theoretical basis for further exploration of the biological function and regulatory mechanisms of the NAC genes of S. muricatum.

Materials and Methods

Identification and biological analysis of SmNAC gene family

The S. muricatum sequencing data were obtained from Illumina HiSeq 2000 sequencer (Accession: SRX1181733 and SRX1177957). Using BMKCloud (https://international.biocloud.net/zh/project/detail/119213). The Trinity assembly of transcripts was carried out by the Eukaryotic Transcriptome Analysis Platform. A total of 68,891 unigenes were obtained, and the N50 was 1,415. The integrity of the assembly was good. We investigated these unigenes further using the NCBI nr (https://www.ncbi.nlm.nih.gov/refseq/about/nonredundantproteins/), Gene Ontology (GO; http://geneontology.org/), Kyoto Encyclopedia of Genes and Genomes (KEGG; https://www.genome.jp/kegg/) and other databases. All NAC genes in S. muricatum were searched. The amino acid sequences corresponding to all the transcripts of S. muricatum were obtained by ORF prediction (http://bioinformatics.ysu.edu/tools/OrfPredictor.html). The amino acid data were uploaded to Pfam (http://pfam.sanger.ac.uk/) for gene function annotation, and finally, a total of 71 NAC nucleotide sequences with NAM and NAC functional domains of S. muricatum were retrieved for further analysis. The NAC protein sequences of Arabidopsis, rice, tomato, potato and pepper were compared with the protein sequences of S. muricatum. All S. muricatum transcript sequence numbers with high similarity to NAC protein sequences (e-value < −50) were identified (Altschul et al., 1997). The online software ExPASy Translate tool (https://web.expasy.org/translate/) was used to translate the SmNAC nucleotide sequences into amino acid sequences, and the Compute pI/MW tool was used on the ExPasy (http://au.expasy.org/tool.html) for analysis to obtain the various physical and chemical properties of the SmNAC proteins, including length, molecular weight, cDNA length, aliphatic index and isoelectric point. The SmNAC proteins were analyzed for conserved domains through the MEME website (http://meme-suite.org), and TBtools was used to draw domain site maps (Chen et al., 2020).

Construction of phylogenetic tree

NAC family protein sequences of Arabidopsis were downloaded from the Plant Transcription Factor Database (http://planttfdb.cbi.pku.edu.cn/), and 24 tomato (S. lycopersicum) NAC genes, 27 potato (S. tuberosum L.) NAC genes and 25 pepper (C. annuum L.) NAC genes were downloaded from the NCBI database (https://www.ncbi.nlm.nih.gov/) for inclusion in the construction of the phylogenetic tree along with the 71 S. muricatum NAC genes. The gene sequence fasta files are shown in Supplemental File 1. MUSCLE (http://www.drive5.com/muscle/) was used to perform multiple alignments of theSolanum muricatum, Arabidopsis, pepper, potato and tomato NAC amino acid sequences. The MEGA 7.0 software and maximum likelihood method (ML, bootstrap = 1,500) were used to construct an unrooted phylogenetic tree (Sudhir et al., 2018) and the online software iTOL (https://itol.embl.de/) was used to annotate the tree.

Tissue-specific expression of SmNAC

Plant materials and drought stress

Detoxified S. muricatum plants planted during the same period were placed in clean water and cultivated for 2 days in a 24 °C constant temperature incubator with 16 h of light and 70% humidity. After this time a sample of leaves from the plants was cut and frozen at −80 °C as a control group for qPCR assay. Then, the plants were transferred into a 15% PEG2000 solution and cultured for 2 days under the same conditions. The roots, stems, leaves and flowers of the plants were cut and frozen at −80 °C, which was used as an experimental group for subsequent qPCR assay. Three independent experiments were conducted.

Fluorescence quantitative analysis of gene expression

The leaves from plants grown without drought stress were used as controls, while the roots, stems, leaves and flowers of plants grown in 15% PEG2000 solution stress for 2 days were subjected to qPCR assay for the 71 SmNAC genes. Primer Premier 6.0 (http://www.premierbiosoft.com/primerdesign/) was used to design the qPCR primers for SmNAC genes. TIANGEN RNAprep Pure polysaccharide polyphenol plant total RNA extraction kit (DP441; TIANGEN, Beijing, China) was used to extract RNA, and TIANGEN miRNA cDNA first-strand synthesis kit (kr211) was used to synthesize the first strand of cDNA. The cDNA of each sample was diluted to 100 ng/µL and used as a template. The APT121 (adenine phosphoribosyl transferase) gene of S. muricatum was used as the internal reference gene. qPCR amplification was performed using the TIANGEN SuperReal PreMix Plus (SYBR Green) (FP205) kit. The total volume of the reaction was 20 µL, containing 10 µL SYBR green supermix, 7.8 µL ddH2O, 0.6 µL for each of the forward and reverse primers and 1.0 µL cDNA. The reaction conditions were 95 °C pre-denaturation for 15 min; 95 °C denaturation for 10 s, annealing for 20 s, and with a total of 45 cycles. The 2−ΔΔCt method was used to calculate the relative expression of the target genes. Omishare Tools heatmap (https://www.omicshare.com/tools/Home/Soft/heatmap) was used to draw the heatmap. A single SmNAC was treated as an object, the expression levels in each tissue were normalized, and each row and column of the heatmap was clustered.

Differential expression of SmNAC genes under different degrees of drought stress

Plant materials and drought stress

Seven-week-old detoxified S. muricatum seedlings were placed in clean water and cultured in a 24 °C constant temperature incubator for 2 days with 16 h of light and 70% humidity, and then transferred to PEG2000 solutions with concentrations of 5%, 10% and 15% for culture under the same conditions. After 0, 4, 8, 12, 18, 24 and 32 h of drought stress, the leaves were frozen at −80 °C. The experiments were repeated thrice.

Fluorescence quantitative analysis of gene expression

Based on the tissue-specific expression of the 71 SmNACs and the analysis of conserved domains, SmNAC3, SmNAC4 and SmNAC8 were selected. These genes had significant expression in the leaves and were comprised of five identical conserved domains (Motifs 13, 20, 6, 15, 3). SmNAC83 with five conserved domains (Motifs 1, 2, 5, 17, 4) was used as a reference, while the clusters SmNAC100 (with Motifs 1, 5 and 17), SmNAC91 (with Motifs 1 and 2) and SmNAC69 (with Motifs 2 and 5) containing sections of the identical domains were selected. The expression level of each gene at 0 h was set as a control, and the S. muricatum APT121 gene was set as an internal reference gene. The qPCR assay was performed for the above seven SmNAC genes. The qPCR experimental method was the same as that in “Fluorescence Quantitative Analysis of Gene Expression”. The seven SmNAC gene-specific primer sequences and their annealing temperatures are shown in Table 1. GraphPad Prism8 was used to draw a histogram representing the levels of gene expression.

Table 1 PCR primer information table.

The information of upstream and downstream primers used for the screening of 7 SmNAC genes and the annealing temperature set during qPCR assay.

Primer name	Sequence (5′-3′)	Annealing temperature (°C)	
SmNAC03-1-F	GGTTAGGTCCATAATAGCA	52	
SmNAC03-1-R	AGCCAATGTCAACAAGAG	
SmNAC04-1-F	CTGCCACTACCATCACCAAG	57	
SmNAC04-1-R	TTCTAGCCGCTCATCTCG	
SmNAC08-1-F	TCAGAATCGTCGGCGTCA	58	
SmNAC08-1-R	TCCTCCCGTTCCATCACA	
SmNAC83-1-F	TTGTAAACTTCAACCTCC	51	
SmNAC83-1-R	TGTATTAAGATTACCACCAG	
SmNAC100-1-F	CAAGCCCTGAAATGTGAA	58	
SmNAC100-1-R	CAGTAAAGAATGATTGGGTGAT	
SmNAC91-1-F	GTATCCTCGTTCTGTTGC	58	
SmNAC91-1-R	GCCTGATTTGTCTGTTGT	
SmNAC69-1-F	TGCAGACCTTGCTCCTAG	56	
SmNAC69-1-R	ATAACTCCGCCTCCACTC	
APRT121-R	GAACCGGAGCAGGTGAAGAA	60	
APRT121-D	GAAGCAATCCCAGCGATACG	

SmNAC amino acid sequence alignment

The seven SmNACs screened in response to drought stress in “Differential Expression of SmNAC Genes Under Different Degrees of Drought Stress” were aligned with the other solanaceous sequences using BLAST-P (https://blast.ncbi.nlm.nih.gov). It was found that the four NAC amino acid sequences, SmNAC83, SmNAC91, SmNAC100 and SmNAC69, showed good alignment with other Solanaceae NAC family members, and had a high degree of similarity to the members of the NAC family of peppers, potatoes, tomatoes and tobacco. The NAC gene sequences of these Solanaceae crops were downloaded and grouped to the same group with the corresponding SmNAC genes. The online software ExPASy Translate tool (https://web.expasy.org/translate) was used to translate the downloaded sequences. Multiple sequence alignment by CLUSTALW (https://www.genome.jp/tools-bin/clustalw) was performed on the four groups of NAC amino acid sequences, and the alignment results were presented using Jalview software (https://www.jalview.org/).

Results

Identification of SmNAC gene family members and protein structural analysis

In this experiment, a total of 71 S. muricatum NAC nucleotide sequences with NAM and NAC functional domains were retrieved for analysis. The analysis of the physical and chemical properties of the SmNAC genes using ExPasy showed that the protein sequences of the different NAC transcription factors were quite different, the amino acid lengths ranged from 64 to 1,201 aa, with an average value of 312.04 aa, while the protein molecular weight was 7.73~139.15 kDa, with an average value of 36.18 kDa (Table 2). Among them, SmNAC63 had the shortest amino acid length, with only 64 aa and SmNAC70 had the longest amino acid length reaching 1,201 aa. SmNAC41 had the highest isoelectric point at 10.83, and SmNAC68 had the lowest isoelectric point at 4.67. Among them, there were 11 NAC proteins with relatively acidic isoelectric points of less than 7. The isoelectric points of the remaining 60 NAC proteins were all greater than 7 and, thus, relatively alkaline. The hydrophilicity of SmNAC16 was the lowest at ~−1.311, and that of SmNAC9 was the highest at ~0.732. Out of the 71 NAC proteins, 40 proteins had a hydrophilicity value of less than 0, which were hydrophilic, while 31 proteins had a hydrophilicity value of greater than 0, making them hydrophobic.

Table 2 The analysis of physical and chemical properties of SmNAC genes.

Structural, physical, and chemical properties of the 71 Solanum muricatum NAC gene family proteins, including the length of cDNA, amino acid length, protein molecular mass, isoelectric point, hydrophilicity, and aliphatic amino acid index of each SmNAC.

Gene name	Number of amino acid	Length of cDNA	Theoretical pI	Molecular weight	Aliphatic index	Grand average of hydropathicity (GRAVY)	
SmNAC001	281	952	9.89	33,076.93	74.91	−0.556	
SmNAC055	72	246	9.89	8,623.95	81.25	−0.804	
SmNAC003	269	839	4.84	29,167.80	81.97	−0.314	
SmNAC004	349	1,060	5.00	38,489.85	91.35	−0.275	
SmNAC005	242	755	9.61	26,710.34	92.73	0.293	
SmNAC006	250	821	9.15	29,081.58	87.76	−0.346	
SmNAC007	202	637	9.84	22,121.79	98.91	0.371	
SmNAC008	292	896	5.17	32,702.08	78.12	−0.454	
SmNAC009	174	547	9.58	20,602.20	139.48	0.732	
SmNAC010	75	263	9.28	9,084.77	78.00	−0.12	
SmNAC011	78	252	9.93	9,676.84	111.03	0.273	
SmNAC012	439	1,393	6.65	51,160.53	81.16	−0.319	
SmNAC013	303	962	9.98	35,144.76	102.54	0.391	
SmNAC083	417	1,267	9.51	48,606.85	79.88	−0.456	
SmNAC015	383	1,228	8.31	44,992.07	103.79	0.173	
SmNAC016	89	274	10.58	9,763.43	38.54	−1.311	
SmNAC017	86	278	10.02	9,683.25	106.51	0.294	
SmNAC018	415	1,338	9.81	48,541.71	113.18	0.342	
SmNAC019	190	603	9.15	22,632.29	108.26	0.038	
SmNAC020	488	1,583	10.32	57,480.91	87.21	−0.384	
SmNAC021	386	1,209	9.92	43,804.80	110.57	0.295	
SmNAC022	834	2,632	10.38	98,282.25	99.46	−0.017	
SmNAC023	264	889	9.63	30,651.24	89.43	−0.209	
SmNAC024	510	1,657	9.77	59,479.00	110.61	0.281	
SmNAC025	467	1,499	9.58	53,875.50	103.45	0.204	
SmNAC026	216	687	9.27	25,484.21	107.87	0.582	
SmNAC027	159	547	9.85	18,850.57	87.04	−0.525	
SmNAC028	299	937	9.97	34,173.24	121.91	0.426	
SmNAC029	432	1,345	10.17	50,313.25	100.09	0.027	
SmNAC030	186	592	9.58	21,963.10	129.84	0.433	
SmNAC031	109	361	9.37	12,902.56	58.17	−1.08	
SmNAC032	405	1,279	10.37	47,851.51	101.98	0.081	
SmNAC033	297	1,043	10.58	34,504.44	66.33	−0.799	
SmNAC034	105	320	8.90	12,591.10	130.76	0.533	
SmNAC035	96	292	9.58	11,396.40	74.06	−0.708	
SmNAC036	497	1,591	10.46	57,968.21	105.92	0.056	
SmNAC037	140	440	10.37	16,809.58	84.21	−0.711	
SmNAC038	139	425	9.62	15,597.49	119.78	0.489	
SmNAC100	379	1,177	10.04	44,033.32	87.18	−0.199	
SmNAC040	517	1,583	10.03	61,149.56	102.34	0.227	
SmNAC041	258	824	10.83	30,624.21	93.64	−0.107	
SmNAC042	102	338	5.41	11,987.43	63.04	−0.689	
SmNAC043	151	471	9.50	16,870.03	116.16	0.604	
SmNAC044	319	969	7.22	37,036.11	76.39	−0.51	
SmNAC045	466	1,550	8.99	53,488.19	88.22	0.001	
SmNAC046	775	2,393	9.16	88,790.28	71.07	−0.655	
SmNAC047	82	255	10.10	9,988.00	124.76	0.157	
SmNAC048	109	357	5.71	12,878.40	52.75	−0.585	
SmNAC091	164	495	6.85	18,780.42	77.74	−0.329	
SmATAF2	233	713	9.62	26,838.87	74.12	−0.551	
SmNAC051	269	844	10.41	31,395.21	106.80	0.205	
SmNAC052	206	728	9.36	24,289.65	73.30	−0.65	
SmNAC053	69	211	8.06	7,808.08	98.70	0.472	
SmNAC054	550	1,722	10.20	63,209.67	119.78	0.486	
SmNAC002	516	1,569	9.29	59,616.37	82.95	−0.169	
SmNAC056	430	1,315	9.34	47,617.09	88.67	−0.193	
SmNAC057	289	879	6.44	33,405.21	71.19	−0.323	
SmNAC058	876	2,748	9.40	101,501.11	87.76	−0.366	
SmNAC059	496	1,546	9.87	58,173.58	91.57	−0.034	
SmNAC060	428	1,384	9.36	49,835.10	88.57	−0.239	
SmNAC061	151	485	10.41	17,769.13	94.24	0.196	
SmNAC062	161	492	10.23	19,508.97	85.90	−0.054	
SmNAC063	64	211	6.02	7,730.94	109.53	0.283	
SmNAC064	805	2,555	9.06	92,552.60	74.48	−0.531	
SmNAC065	81	258	9.60	9,069.87	108.02	−0.007	
SmNAC066	306	1,055	9.92	36,121.10	98.50	0.241	
SmNAC067	297	985	9.88	34,929.32	85.69	−0.179	
SmNAC068	85	292	4.67	10,191.52	104.35	−0.466	
SmNAC069	617	1,855	8.72	70,549.22	69.71	−0.649	
SmNAC070	1201	3,823	9.75	139,153.20	107.37	0.33	
SmNAC071	68	225	6.90	8,011.07	81.76	−0.475	

Conserved domain analysis of SmNAC gene family

Conserved domain analysis was performed on the entire sequences of the 71 S. muricatum NAC proteins. Forty-four SmNAC protein sequences were identified as having conserved domains (Motifs) (Fig. 1). These conserved domains comprised 20 Motif categories (Fig. 2). It can be seen from Fig. 2 that Motifs 4, 12 and 8 had the highest frequency and appeared 10 times during the search. We observed that SmNAC3, SmNAC4 and SmNAC8 had only five identical motifs. The relative positions of these motifs were fixed and the distance was relatively close. This indicates conservation and suggests that the biological functions of SmNAC3, SmNAC4 and SmNAC8 are likely to be similar. Motifs 10, 7, 12 and Motif 8 appeared together in five SmNAC protein sequences. The relative position of the above motif was constant and close to the C-terminus. Motifs 1, 2, 5 and Motif 17 were found together in four SmNAC protein sequences simultaneously, two of which had only four motifs. The relative positions of these four motifs were constant and located near the N-terminus.

Figure 1 The position of the 44 conserved domains of SmNAC protein on the amino acid chain.

The position of the 44 conserved domains of SmNAC protein on the amino acid chain. The rectangles of different colors represent different motifs and their positions on each of the NAC sequences. The correspondence between the rectangles of each color and motif is shown in the key in the figure. The horizontal axis at the bottom of the figure is the reference axis of the length of the amino acid chain.

Figure 2 The 20 conserved motifs (Motifs A–T) identified in Solanum muricatum NAC proteins.

The 20 conserved motifs (Motifs A–T) identified in Solanum muricatum NAC proteins. The length, degree of conservation, and the sharing or substitution of every amino acid in each of the motifs are displayed. Each motif had a corresponding coordinate axis, the abscissa represents the corresponding position of each amino acid on the motif, and the ordinate represents sections.

Phylogenetic analysis of SmNAC gene family

To better understand the phylogenetic relationships within the SmNAC gene family, the NAC protein sequences of S. muricatum (71 NAC protein sequences), Arabidopsis (30 NAC protein sequences), tomato (24 NAC protein sequences), potato (27 NAC protein sequences) and pepper (25 protein sequences) were subjected to multiple alignments to construct an unrooted evolutionary tree (Fig. 3). S. muricatum is the target species of this experiment, so only the S. muricatum were divided into categories. According to their branch on the tree, the 71 SmNAC transcription factors were roughly divided into seven categories: Classes I, II, III, IV, V, VI and VII, of which Classes II and III had only a single NAC gene, while Class II had 15 S. muricatum NAC genes and Class III had 20 S. muricatum NAC genes. These two major categories of S. muricatum genes had no obvious homology with sequences from other species. Class I contained 11 S. muricatum and 1 tomato NAC sequence, with these sequences showing relatively high similarity. There were 17 S. muricatum NAC and 1 pepper NAC distributed in Class IV. Classes VI and VII branched from the pepper NACs, which formed the main branch, containing 2 and 1 S. muricatum NAC, respectively, indicating that S. muricatum had a closer evolutionary relationship with pepper than with Arabidopsis and other solanaceous crops. SmNAC022 and SmNAC064 were found on independent branches. Class V was the largest branch in the evolutionary tree with 41 NACs, of which only 3 belonged to S. muricatum. This branch included all species participating in the alignment; a total of 4 Solanaceae NACs, including SmNAC045, SlNACMTF2, CaNAC091 and StNAC078. These further formed a sub-branch of Class V, with obvious homology in evolution and functional similarity.

Figure 3 Unrooted phylogenetic tree showing the relationship between the NAC family members of Arabidopsis, pepper, tomato, potato, and Solanum muricatum.

Unrooted phylogenetic tree showing the relationship between the NAC family members of Arabidopsis, pepper, tomato, potato and Solanum muricatum. Each species is represented by a background color as shown in the key in the figure. Classes of different colors represented different subfamilies of Solanum muricatum NAC.

Effect of drought stress on the seedlings of S. muricatum

To further understand the effects of drought stress on S. muricatum seedlings, this experiment used three concentrations (5%, 10% and 15%) of PEG to simulate different degrees of drought stress, and observed the changes in S. muricatum seedlings under different degrees of drought stress (Fig. 4). The results showed that under drought stress, the seedling leaves wilted and showed chlorosis, which appeared earlier in seedlings grown in higher PEG concentrations that is, conditions of greater drought stress. Under mild drought stress (5% PEG solution), the old leaves of the seedlings sagged significantly after 18 h with no obvious change in the whole plant from 18 to 32 h. Most of the leaves remained upright after 32 h of stress induction. Under moderate drought stress (10% PEG solution), the leaves sagged around 12 h of exposure; the old leaves on the lower half of the seedlings wilted and curled after 18 h, with an increased degree of wilting with time and resulting in complete chlorosis at around 32 h. The leaves in the lower half wilted completely, while the leaves in the upper half remained unaffected. Under severe drought stress (15% PEG solution), the older leaves on the lower half of the seedlings wilted and sagged after 12 h while the top leaves began to wilt at around 18 h of exposure. The curl of the lower half of the leaves aggravated around 24 h with no upright leaves on the plant and chlorosis appeared; by 32 h the plants appeared completely wilted with black and shrunken leaves.

Figure 4 Changes in plant shape of Solanum muricatum seedlings with 7 weeks old after drought stress.

This figure shows the changes in plant shape of Solanum muricatum seedlings with 7 weeks old after drought stress. The drought stress experiment was conducted in an incubator at 24 °C with 70% relative air humidity, and 16 h/d light. The red gradient deepening process indicated the passage of drought stress time, and the green gradient deepening process indicated the increase of PEG solution concentration. The photos show the morphological changes of Solanum muricatum plants at various sampling time points (4, 8, 12, 18, 24 and 32 h) and in the presence of PEG at three different concentrations (5%, 10% and 15%).

The tissue-specific expression of SmNAC genes

To analyze the function of S. muricatum NAC genes under drought stress, qPCR was used to analyze the expression of the 71 NAC genes in four different tissues and organs, namely, roots, stems, leaves and flowers. The differences in the expression of individual SmNAC in each tissue were analyzed and each row was normalized. The results showed that there were significant differences in the expression levels of different NAC genes in different tissues and organs of S. muricatum (Fig. 5). A total of 13 SmNAC genes were not expressed in the above four tissues, indicating that they did not respond to drought stress, but may respond to other abiotic stresses or biological stresses. According to the normalization and clustering results of each row, the SmNAC gene expression in the root was more characteristic than that in other tissues. The number of genes with the highest or lowest expression in each tissue in the root was more than that of flowers, leaves and stems. The polarization was very obvious, that was, the amplitude of gene up-regulation or down-regulation was higher than that of the other tissues. The number of SmNAC with the highest expression levels was relatively high in flowers, which was second only to the roots, and the genes having an average expression level were maximum in all the tissues. Only the expression level of SmNAC52 in flowers was significantly lower than in other tissues. The clustering results of each column showed that the expression status of SmNAC in leaves and stems was almost similar, and the number of genes at the highest expression level or the lowest expression level was approximately equal. It is worth mentioning that SmNAC3, SmNAC4 and SmNAC8 had five identical conserved domains and their expression in leaves was very significant. It can be speculated that these three genes played a significant role during drought stress in S. muricatum plants, and they may have similar biological functions.

Figure 5 Heat map.

The Heat map shows the expression of Solanum muricatum NAC genes in four tissues of root, stem, leaf and flower under drought stress. A positive value meant higher than the average level, 0 was the average level, and a negative value represents below-average levels.

Gene expression trend of SmNAC under different degrees of drought stress

Figure 4 shows the real-object diagram of three kinds of PEG concentration simulating drought stress. With the increase of the PEG solution concentration, the rate of wilting and chlorosis of S. muricatum plants was accelerated. Based on the tissue-specific expression and conserved domain analysis of the 71 SmNACs, seven SmNACs were selected for qPCR analysis to further explore the changes in SmNAC gene expression over time under different concentrations of PEG stress. As a general observation, as the of drought increased, expression (both upregulated and downregulated) of the SmNAC genes increased, reaching extreme values more rapidly (Fig. 6). A gene expression level of ≥2 or ≤0.5 was regarded as a significant change in the gene expression level. The expression trends of SmNAC3, SmNAC4 and SmNAC8 were similar, and the expression curve showed a single peak curve. It is possible that these three proteins have similar biological functions. Among them, the expression levels of SmNAC3 was the highest among all the tested genes, reaching 46 times under mild drought stress and 135 times under severe drought stress. During the experimental sampling period, SmNAC8 had consistently higher expression levels under severe drought stress and the relative change was smaller than that for the other genes. The expression trends of SmNAC83, SmNAC100 and SmNAC91 were similar, and the expression curve was roughly bimodal, all reaching the first peak at 4 h, then showing a downward trend and decreasing to the minimum at 18 h or 24 h, with gradual increase thereafter. Among them, SmNAC100 responded to drought stress most significantly. The expression level under mild drought stress reached 45 times at 4 h, and the expression level under severe drought stress reached 122 times. The relative expression levels of SmNAC83 and SmNAC91 between 18 and 24 h were around 1, which can be regarded as the normal expression level, while SmNAC69 was a down-regulated gene. With the increase in the degree of drought stress, the minimum value was reached more rapidly, at 8 h under severe drought stress, and at 18 h under mild drought stress, with downregulation of the gene at almost 1/10 of normal levels. The expression of SmNAC69 was significantly suppressed.

Figure 6 Changes in the expression of the selected (A–G) Solanum muricatum NAC genes over time under different degrees of drought stress.

The histogram shows the changes in the expression of the selected (A–G) Solanum muricatum NAC genes over time under different degrees of drought stress. Three kinds of PEG solutions with 5%, 10% and 15% concentrations were used to stress the Solanum muricatum plants respectively. The leaves of the plants were taken after 0, 4, 8, 12, 18, 24 and 32, and the expressions of Solanum muricatum NAC gene were detected by qPCR. The 2-ΔΔCT method was used to calculate the relative expression of the target genes. The abscissa represents each sampling time point of the experiment (4, 8, 12, 18, 24 and 32 h). The ordinate indicates the relative expression levels. White, gray and black represent 5%, 10% and 15% PEG stress experimental groups, respectively.

NAC amino acid sequence alignment of various species

The amino acid BLAST alignment results of the seven SmNACs (Fig. 6) that actively responded to drought stress were analyzed, and it was found that four NACs that is, SmNAC83, SmNAC100, SmNAC91 and SmNAC69 showed relatively good alignment with the Solanaceae crops. The alignment results all included the same five species (Solanum pennellii, S. tuberosum, C. annuum, N. tabacum and N. tomentosiformis) (Fig. 7). Overall, these SmNAC showed strong homology with tomato, potato, pepper and tobacco sequences. Tobacco crops showed strong specificity in the process of comparison. The amino acid comparison results showed that SmNAC69 was highly similar to tomato, potato and pepper. The tobacco sequence, at 140 aa, had an additional two residues, resulting in a subsequent conservative conserved sequence shift; SmNAC83 had the highest similarity with tomato, and, among the fragments, the similarity of the sequences from the five plants in the range of 1–105 aa was high while different numbers of glutamine residues were seen in the different plant sequences beyond residue 105. SmNAC91 had the highest similarity with tomato and potato, and tobacco and pepper showed strong high specificity in this comparison. It is worth mentioning that the specificity of capsicum was the most prominent; the alignment of NAC100 was the best among the four groups of genes. Apart from the substitution of isolated amino acids residues, the sequences showed a high degree of conservation.

Figure 7 Multiple sequence alignments of four groups of NAC genes.

Multiple sequence alignments of (A)–(D) groups of NAC genes. The Zappo Colours color scheme (http://www.jalview.org/version118/documentation.html#zappo) was used to present the amino acid sequence. Conservation: Conservation of total alignment of less than 25% gaps. Quality: Alignment Quality based on Blosum62 scores. Consensus: PID (http://www.jalview.org/version118/documentation.html#pid). Occupancy: Number of aligned positions.

Discussion

Bioinformatics analysis of S. muricatum NAC gene family

NAC transcription factors are generally less than 400 amino acids long with highly conserved N-termini. The protein sequence contains the approximately 150-residue NAC domain which is composed of five substructures: A, B, C, D and E (Hisako et al., 2003). The lengths and sequences of the C-terminal regions are highly diverse (Xin-Jian et al., 2005). In this study, it was found that the N-terminus of SmNAC2, SmNAC44, SmNAC83 and SmNAC46 contained Motifs 2, 1, 5 and 17, while the C-terminus of SmNAC83 had Motif 4, and the SmNAC had Motifs 1, 5 and 7, which is consistent with the characteristic NAC features of a highly conserved N-terminus and a diversified C-terminus (Yujie et al., 2008). However, there were certain differences in the N-terminal conserved domains. For example, SmNAC3, SmNAC4 and SmNAC8 had five identical domains at the N-terminus (Motifs 13, 20, 6, 15 and 3), but they did not share the same Motifs with SmNAC2 and the other above-mentioned genes. ATAF1 and 2 is a typical NAC transcription factor identified in Arabidopsis, with a highly conserved NAC domain at the N-terminus (Swati et al., 2012). The SmATAF2 identified in this study had the same Motif 2, 1 and 5 at the N-terminus as SmNAC2. It also had the same Motif 20 as SmNAC3 and other sets of genes. It has also been reported that the N-terminal domains of the Solanaceae pepper and potato NAC families are not completely consistent (Schafleitner et al., 2013; Weiping et al., 2018). These findings indicate that the N-termini of NAC transcription factors are not completely conserved.

The phylogenetic tree showed that the 71 members of the NAC family of S. muricatum can be divided into seven subfamilies which are closely related to the NACs of pepper in the Solanaceae family. From previous studies on the evolution of the NAC family, it has been found that there is an evolutionary relationship between Arabidopsis and the solanaceous crops such as peppers, potatoes and tomatoes (Tweneboah & Oh, 2017). Class V in the evolutionary tree constructed in this study also supports this phenomenon. In the evolutionary analysis of the pepper NAC family, it was found that there was a branch containing only pepper NAC (Schafleitner et al., 2013), and a similar situation was observed while constructing the Class II and Class III evolutionary tree. This S. muricatum NAC does not seem to have a close relationship with NACs other species, which suggests that they may play a unique role in the growth and development of S. muricatum. NACMTFs are unique transmembrane proteins in plants. After receivingstress signals, they are transferred to the plasma membrane and truncated at the C-terminus for translocation either into the cytoplasm or nucleus, thereby completing the transmembrane signal transmission. This pathway has been reported in Arabidopsis and other crops (Kim, Kim & Park, 2007). In potato and tomato, which also fall in the Solanaceae family, several NACMTFs have been identified (Bhattacharjee et al., 2017; Schafleitner et al., 2013). SINACMTFs are tomato-specific membrane-binding factors, and in this study, it was found that SmNAC45 and SmNAC40 were closely related to SINACMTF2 and SINACMTF3, suggesting that they may have similar biological functions. It is worth mentioning that SINACMTFs will significantly affect the activity of the NAC promoter. The same SlNACMTF shows different regulatory modes for different promoters (Bhattacharjee et al., 2017), but whether SmNAC40 and SmNAC45 follow the same regulatory pathway needs further research.

Expression of S. muricatum NAC transcription factors under drought stress

According to the research by Qinqin et al. (2012) the expression levels of the SlNAC3 gene in the flowers and roots of tomato were relatively high, indicating that the growth of flowers and roots was under the influence of this transcription factor. Also, the expression of SmNAC9, SmNAC55, SmNAC25, SmNAC59 and SlNAC3 was similar, which indicates that they may be closely related to the growth and development of flowers and roots. It is worth mentioning that SlNAC3 has a diversity of functions. Salt stress, drought stress, and ABA treatment can inhibit the expression of SlNAC3. The analysis results showed that SlNAC3 may interact with environmental and endogenous stimuli and act as a transcriptional activator in plants through the ABA signaling pathway and play a role in response to salt and drought stress in plants (Qinqin et al., 2012). The functional prediction and verification of SlNAC3 can provide research directions for future research on the functions of the SmNAC genes.

In this study, while investigating the tissue-specific expression of SmNAC under drought stress, we found that, compared with other tissues, the highest expression of the upregulated SmNACs was in the roots, followed by the flowers, while most of the genes showed average expression levels. In general, the expression levels were significantly upregulated, consistent with the hypothesis that flowers are the main source of water loss during drought stress compared to the stems and leaves (Ibrahim et al., 2020). There were many genes showing the highest and lowest expression levels in the roots, where significant polarization was observed, in other words, the amplitude of the gene upregulation or downregulation was greater than in other tissues, with SmNAC23, SmNAC40, and others being typical representatives. In potato, the expression of StNAC103 in the roots was significantly higher than that of other genes (Schafleitner et al., 2013); StNAC103 has been shown to be a repressor of suberin polyester and suberin-associated wax deposition (Soler et al., 2020). Whether SmNAC23 and other genes also have related functions needs to be further confirmed. In research on the potato NAC family, it was found that StNAC33, StNAC55 and StNAC97, which had seven identical motifs and are closely related, had similar expression trends in tissue-specific expression (Schafleitner et al., 2013). Similar findings were found in the tissue-specific expression of SmNACs 3, 4 and 8 with five identical motifs and all belonging to Class III, which showed greater expression in the leaves compared to the other tissues. It is speculated that these three SmNACs with the same motif may execute the same biological function in the leaves.

Regulation of NAC transcription factors under abiotic stress

In recent years, there has been much research on the NAC gene family. Many species have been used for the identification and functional prediction of the NAC gene family. The latest research on Moso bamboo (Phyllostachys edulis) identified 94 NAC genes, of which 15 PeNACs played a significant role in secondary cell wall biosynthesis and lignification of the bamboo (Xuemeng et al., 2019). MdNAC52 and MdNAC42 in apple (Malus domestica) can regulate the synthesis and accumulation of anthocyanins (Shuangyi et al., 2020; Sun et al., 2019b). Likewise, 183 NAC genes were identified in the white pear (Pyrus bretschneideri), which were divided into 33 subgroups, out of which the C2f, C72b and C100a subgroups actively responded to drought and low-temperature stress (Xin et al., 2019). With increased research on the biological functions of NAC transcription factors, it has been shown that NAC transcription factors can regulate a wide range of plant life processes including plant growth and development, the growth and senescence of the cell wall and root through plant hormone signaling pathways (Wei et al., 2018; Yongil et al., 2019), and improving plant resistance by responding to a variety of abiotic and biological stress factors (Du et al., 2020; Jia et al., 2019). In terms of abiotic regulation, the research on the pathways and mechanisms of NAC family genes for abiotic stress has been intensively studied. For example, the NAC11 gene of Ammopiptanthus mongolicus encodes a transcription factor that can respond to drought, low temperature, high salt and other stresses. The translated protein is localized in the nucleus and plays an important role in drought, cold and salt tolerance (Xinyue et al., 2019). In the NAC family of corn (Zea mays L.), overexpression of ZmNAC 071 in Arabidopsis enhanced the sensitivity of transgenic plants to ABA and osmotic stress (Lin et al., 2019). Meanwhile, studies on splicing variants of the NAC family have been carried out in corn (Zhang et al., 2019). In the Solanaceae family to which the S. muricatum belongs, CaNAC064 of the pepper NAC family interacts with low-temperature-induced plant cell single-chain protease proteins to positively regulate cold tolerance in plants. Analysis of the protein structure and expression pattern of NtNAC2 in tobacco has been completed (Han et al., 2014). The research of these closely related species has provided a good foundation for the follow-up research of the S. muricatum NAC family.

Although previous studies reported the association of the NAC gene family with abiotic stress, our experiments are a more detailed exploration. Here we investigated the expression trends and functional prediction of the NAC genes of S. muricatum under drought stress. The results showed that the expression levels of SmNAC83 and SmNAC91 initially increased, then fell back to normal levels before increasing again. This is very similar to the reported expression trend of SlNAC8 in the tomato NAC family under PEG drought stress where increases were associated with higher degrees of stress (Kou et al., 2017). The expression level of SmNAC69 first showed a decreasing trend and then gradually recovered to the normal expression level, which is similar to the reported expression trend of SlNAC9 under PEG drought stress (Kou et al., 2017). Subsequent studies in the tomato NAC family have confirmed that the SlNAC9 gene is involved in carotene synthesis and fruit softening of tomato (Kou et al., 2018). Whether SmNAC69 has the same biological function is still unknown and hence it is necessary to verify the specific function and mechanism of this gene in future research using either transgenic or gene silencing technology. In the genome-wide analysis of the pepper NAC transcription factor gene family, CaNAC showed three expression trends under drought stress. There are upregulated genes such as CaNAC23, CaNAC53, CaNAC37 that exhibit a single peak curve, upregulated genes such as CaNAC72 and CaNAC79 that exhibit a double peak curve, as well as downregulated genes such as CaNAC56, CaNAC33, CaNAC35 (Weiping et al., 2018). The seven SmNACs selected in this study also had three expression trends that is, SmNAC3, SmNAC4 and SmNAC8 are single-peak upregulated genes, SmNAC83, SmNAC100 and SmNAC91 are double-peak upregulated genes, and SmNAC69 is a downregulated gene. This similar expression trend indicates that they may also have similarities in their functions. In addition to the results of evolutionary tree analysis, this also provides evidence for the strong homology between SmNAC and CaNAC. It is worth mentioning that both CaNAC35 and SmNAC69 are downregulated genes and the expression trend is almost the same. CaNAC35 has been shown to be a positive regulator of tolerance in response to abiotic stress in pepper and functions through multiple signaling pathways (Huafeng et al., 2020). This can help in providing research directions for functional verification of SmNAC69 in the future.

NAC amino acid sequence alignment of related species

In general, SmNAC has high homology with the S. pennellii, S. tuberosum, C. annuum, N. tabacum and N. tomentosiformis NAC families with the highest degree of similarity between SmNAC and SpNAC. The results showed that orthologous NACs in each species were conserved with small differences with the differences caused by subtle alterations such as the addition or replacement of individual amino acids. SmNAC100, SpNAC100, StNAC100, CaNAC100, NtNAC100 and NtoNAC100 have high degrees of similarity, which may be derived from the same ancestral NAC gene. As SmNAC100 is the most similar gene to Nicotiana NAC among the four SmNAC genes, and tobacco is also a popular crop for functional verification of NAC genes (Sun et al., 2019a; Wang et al., 2020), we will use SmNAC100 as the preferred gene for functional verification in tobacco crops in the next step. The functional verification of NAC family genes in Solanaceae, which are more closely related to SmNAC such as C. annuum NAC1 and S. lycopersicum NAC2, also provides theoretical guidance for the further study of SmNAC functions (Borgohain et al., 2019; Xi-Man et al., 2020). In this study, the role of NAC genes in ginseng fruit under drought stress was preliminarily discussed and compared with Solanaceae crops. However, the functional verification of these genes needs to be further investigated in future research.

Conclusions

Using genome-wide identification of the S. muricatum NAC transcription factor family, we identified a total of 71 family members belonging to seven subfamilies. These proteins had typical NAC conserved domains and their gene expression was significantly influenced by drought stress. The expression levels were relatively high in roots and flowers while expression in the leaves and stems was very similar. The expression levels of certain genes in leaves showed three different trends: single peak upregulated expression, double peak upregulated expression and downregulated expression. The Solanaceae NAC amino acid sequence alignment showed a high degree of homology, and S. muricatum NACs were strongly homologous with NACs from pepper, tomato, potato, tobacco and other Solanaceae crops.

Supplemental Information

Supplemental Information 1 RT-PCR heatmap data (Figure 5).

Click here for additional data file.

Supplemental Information 2 Figure 6 RT-PCR data.

Click here for additional data file.

Supplemental Information 3 Primers of heatmap.

Click here for additional data file.

Supplemental Information 4 Protein sequences.

Protein sequences used for the construction of rootless evolutionary tree, identification of conserved domains and analysis of physicochemical properties.

Click here for additional data file.

Supplemental Information 5 Figure 7 protein sequence.

Click here for additional data file.

Supplemental Information 6 Go.

Gene annotation and Pfam annotation.

Click here for additional data file.

Additional Information and Declarations

Competing Interests

Author Contributions

Data Availability

The authors declare no conflict of interest.

Shipeng Yang conceived and designed the experiments, authored or reviewed drafts of the paper, supervision, Project administration, Funding acquisition, and approved the final draft.

Haodong Zhu conceived and designed the experiments, performed the experiments, analyzed the data, prepared figures and/or tables, authored or reviewed drafts of the paper, and approved the final draft.

Liping Huang analyzed the data, authored or reviewed drafts of the paper, and approved the final draft.

Guangnan Zhang analyzed the data, authored or reviewed drafts of the paper, investigation, and approved the final draft.

Lihui Wang analyzed the data, authored or reviewed drafts of the paper, supervision, and approved the final draft.

Xiaoting Jiang analyzed the data, authored or reviewed drafts of the paper, and approved the final draft.

Qiwen Zhong conceived and designed the experiments, authored or reviewed drafts of the paper, project administration, Funding acquisition, and approved the final draft.

The following information was supplied regarding data availability:

The Solanum muricatum sequencing data is available at the SRA database: SRX1181733 and SRX1177957.

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
