# Peer review of "Transcriptome-wide and expression analysis of the NAC gene family in pepino (Solanum muricatum) during drought stress"

_PeerJ, doi:10.7717/peerj.10966_

## Round 0.1 · original submission · Major Revisions

Both reviewers found that the paper provided useful information on the NAC gene family in pepino species and had potential to assist future detailed studies on drought resistance in these species. However, the reviewers made many suggestions for revision to improve the precision of the description of the data and to make the analysis more accessible to readers. Reviewer 1 makes many important suggestions on improving the presentation of the phylogenetic analysis. Also, this reviewer raises the issue that in figure 7 stop codons appear within the protein sequences, raising the possibility that some of the sequences used are pseudogenes. The reviewer points out that you should also provide the nuclear sequences used to predict the protein sequences. Reviewer 2 points out that as you used transcriptome data the title should be changed and makes suggestions on additional tools that should be used for the analyses. Both reviewers ask that the English language be improved. In any revision please take account of all the suggestions of the reviewers and provide a detailed point by point description of the revisions you have made to answer their points.

Reviewer 1 ·

Basic reporting

Just providing the comments for improvement

Language:
English can be improved a bit throughout the manuscript.


Figure Specific comments:

Figures should be improved particularly Figure 1 and 7. Both of which are not sharp.

Figure 1 has the additional issue of requiring a better legend.
The description of the motif figure (sequence logo) is not completely clear. The X-axis description is ok but the Y-axis is not the number of NACs with that motif. The Y-axis provides bits (see Crooks et al. 2004).

Figure 2. Can be improve if the sequences were clustered by their motif content

Figure 3. In phylogeny we typically speak of Unrooted and not rootless trees. Highlighting of the tips does not cover the full name.

Line 337: I suppose that Figure 5 in this sentences refers to Figure 6?


This paper has no clearly stated question as its aim was to describe the NAC gene family in pepino and analysis there expression pattern in during Drought treatment. It is a descriptive paper

Experimental design

Issues with the methods

Line: 118. I have two concerns here: First, the authors mention to have obtained sequencing data from the accessions SRX1181733, SRX1177957 of which only the second accession corresponds to Solanum maricatum. I suppose the authors have taken the transcriptome data from Harraiz et al. 2016. If not, the authors should clearly state how the transcriptome was generated as the SRA accessions provide reads and not transcripts ready for translation.

Line 125: I suppose BLAST was used for comparison of the sequences given that E-value is mentioned in the sentence. If so, please correctly cite the BLAST manuscript.

Line 131: I reckon that MEME was used for MOTIF identification and conserved domains (which was done with PFAM).

Line 142: It is of upmost importance to state which substitution model was used for the maximum likelihood tree estimation. It is also important (although to lesser extend) describe why the model was used. Another concern is whether the authors made any attempt of removing lowly conserved regions from the alignment with for instance TrimAL (Capella-Gutierrez et al. 2009) prior to constructing the tree.

Line 339: What exactly was the criteria (in domains and expression) for selecting the 7 genes for followup. In the discussion (line 508) this is a bit elaborated but still not clearly.


Figure 3
I can understand that this would happen for Arabidopsis in the tree. But I wonder why they don’t mix so much with the other solanum species. The other solanum species do seem to have some specific groups but not as large as pepino. Could this be the effect of using a de novo transcriptome which might generate many transcript isofomrs for the same gene thus blowing into what would appear large species specific groups.

Figure 7. It is not easy to see however, there appear to be stop-codons in the alignment. I confirmed that this is case by directly translating the NAC91 sequecne of Capsicum that is provided in the fasta files of the authors and found the FSSFLFLFSKD*ILAMAVLHEGNQLA in the alignment. Therefore, I am sorry to say that Figure 7 cannot be used to make conclusions

Validity of the findings

Given that stop codons were present in the alignment of Figure 7. I am worried about the protein sequences underlying the e.g. MOTIF , PFAM and Phylogentic analyses which are based on protein sequences. I am sorry to be negative about this point it is crucial that the protein sequences are well predicted. It is therefore important to have provided the protein sequence translations and not the whole gene/cdna sequences (not even CDS). The conclusions/findings of this paper are for a large part dependent on these inferred protein sequences.

Reviewer 2 ·

Basic reporting

No comment.

Experimental design

No comment.

Validity of the findings

No comment.

Additional comments

The manuscript “Genome-wide and expression analysis of NAC gene family in pepino (Solanum muricatum) during drought stress” reports basic information of the NAC gene family in pepino. Most of the information is retrieved using bioinformatic tools, and the expression of NAC genes in different tissues and under different drought stress using qPCR. This work will fundamentally benefit further studies of pepino drought tolerant genes in future. Suggestions for improvement on this manuscript were as follows:
1. The English language should be improved. I suggest that you obtain assistance from a colleague who is well-versed in English or whose native language is English.
2. The NAC sequences were obtained from transcriptome database rather than genome database, so the “Genome-wide” in the title should be replaced by “Transcriptome-wide”.
3. Line 121-123. The authors could consider building model using previously characterized NAC sequences and sequences from other species and perform further searches to verify if any genes are missed. The identified sequences need to be further analyzed to confirm the presence of NAM or NAC domain using other tools, such as InterPro program (http://www.ebi.ac.uk/interpro/), CDD (https://www.ncbi.nlm.nih.gov/Structure/cdd/wrpsb.cgi) or SMART (http://smart.embl-heidelberg.de/).
4. Line 135-139, 258-261. Why were these NAC genes selected in these species? Whether the representativeness of these genes in different subfamilies is considered to better classify SmNAC genes?
5. Line 151-152. Why the samples used for tissue-specific expression undergo drought treatment rather than normal growth?
6. Line 159-160. Please provide the primer information of these 71 SmNAC genes for qPCR.
7. Other comments
Line 17. “(Pepino)” should be placed after “Solanum muricatum” in Line 15.
Line 47. “Solanum muricatum” should be italic.
Line 66, 68 and 336. Pay attention to the correct writing of superscript and subscript in “ddH2O” and “2-ΔΔCT”.
Line 250. Please check that the number 47 is correct.
Line 333. The word “meant” should be “means”.
Line 337. Please check if Figure 5 should be Figure 4.

---

## Round 0.2 · Minor Revisions

The reviewers agreed that your manuscript was much improved, and that it is now almost ready for acceptance. However, reviewer 1 comments that your conclusions would be improved if you looked again at the alignment to maximize the conservation. Could you please consider this point in a minor revision? This is an opportunity to further improve your paper, after this minor revision it will not be sent to reviewers again.

Reviewer 1 ·

Basic reporting

I will go over the main points that I mentioned last time

The English has indeed improved and I could understand the manuscript very fast.

The issues I had wit all figures has been solved except I still have an issue with Figure 7. Although the stop-codon problem has been resolved the alignment can be improved and result in better conservation values. If you look closely you can for instance at 2 gaps at position 140 for all the sequences except the ones of Nicotiana. This would already increase the number of conserved positions. It seems overall that some gaps could be introduced. It is a bit suspicious that the alignment has no gaps at. Please revise the alignment.

The authors recognised that the paper is more descriptive and I am satisfied with their adjustments on this issue.

Experimental design

Concerning the selected model for Phylogeny: Although I am still always in favour of performing model selection. I will accept in this case usage of the GTR as the authors are using it informed based on literature.

Validity of the findings

Given the responses of the Authors in the Rebuttal letter and the major modifications to the text (including my concerns) I am satisfied. The only thing I would urge the authors to do is to improve the alignment in Figure 7 because it will so a better conservation.

Additional comments

Once again, I am happy with the changes made by the Author and I accept the manuscript.

---

## Round 0.3 · Minor Revisions

Thank you for dealing with these minor revisions. The additional point below was raised during editorial assessment.

An important aspect of the manuscript was to characterize the 71 NAC genes. The sequence data was presented and some features were associated with the expression of the genes under a range of conditions. It was mentioned that annotations were used to classify the sequences, and as expression profiles were tested, it would be needed to add the appropriate annotations to the sequences. This potentially can be done in the supplemental files, creating a summary table detailing the unique features uncovered for select NAC genes and GO: terms added in numerical and textual contexts.

Journal manuscripts are often scanned by text-mining software that locates and extracts core data elements, like gene function. Adding standard ontology terms, such as the Gene Ontology (GO, geneontology.org) or others from the OBO foundry (obofoundry.org) can enhance the recognition of your contribution and description. This will also make human curation of literature easier and more accurate. It was mentioned, but none of this information was visible.

The manuscript was well presented and clear. The missing factors would be the summary tables with annotations to detail developmental, tissue, and treatment conditional expression; best done using GO terms. I consider this manuscript requiring minor revisions as discussed above.

---

## Round 0.4 · accepted · Accept

Thank you for the second revision, I am pleased now to accept your manuscript.

The Section Editor added the following comments that should be considered as you prepare the submission fo publication:

"The suggested additions have been made. The references in the manuscript pointing to the following files were not seen; however, once identified or added in the final revision the manuscript should be ready. The line 142 mentions the supplementary FASTA file data, but the supplemental files may not be labelled properly; the files peerj-49879-All_protein_sequence.txt needs a better title. The other FASTA file for Figure 7 needs to be accordinly be adjusted (peerj-49879-The_protein_sequence_of_Fig._7.txt). It may be better to just add a statement saying there is supplemental data, and better define the title of each file provided. There were several modifications made from the previous revision, but they read clearly. Other edits are included below: EDITS LINE NO: / BEFORE / AFTER / [COMMENTS] LINE 21: / family of contains / family contains / [.] LINE 30: / significant and significant / significant / [.] LINE 57: / west Gansu / West Gansu / [.] LINE 63: / osmotic stress, that, / osmotic stress; that, / [.] LINE 66: / tolerance act as / tolerance which act as / [.]"